# Normal-Abnormal Decoupling Memory for Medical Report Generation

**Guosheng Zhao[1], Yan Yan[2], Zijian Zhao[1]***

[1]School of Control Science and Engineering, Shandong University
[2]Department of Computer Science, Illinois Institute of Technology.
{zhgs,zhaozijian}@sdu.edu.cn, yyan34@iit.edu

## Abstract

The automatic generation of medical reports plays a crucial role in clinical automation. In contrast to natural images, radiological images exhibit a high degree of similarity, while medical data are prone to data bias and complex noise, posing challenges for existing methods in capturing nuanced visual information. To address these challenges, we introduce a novel normal-abnormal semantic decoupling network that utilizes abnormal pattern memory. Different from directly optimizing the network using medical reports, we optimize visual extraction through the extraction of abnormal semantics from the reports. Moreover, we independently learn normal semantics based on abnormal semantics, ensuring that the optimization of the visual network remains unaffected by normal semantics learning. Then, we divided the words in the report into four parts: normal/abnormal sentences and normal/abnormal semantics, optimizing the network with distinct weights for each partition. The two semantic components, along with visual information, are seamlessly integrated to facilitate the generation of precise and coherent reports. This approach mitigates the impact of noisy normal semantics and reports. Moreover, we develop a novel encoder for abnormal pattern memory, which improves the network's ability to detect anomalies by capturing and embedding the abnormal patterns of images in the visual encoder. This approach demonstrates excellent performance on the benchmark MIMIC-CXR, surpassing the current state-of-the-art methods.[1].

## 1 Introduction

Report writing is a critical responsibility for radiologists. Automatic report generation is clinically significant as it alleviates the workload of physicians. Recently, substantial advancements have

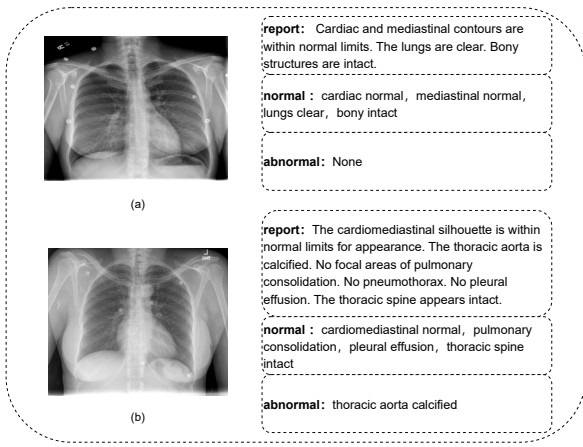

Figure 1: Radiological images, medical reports and normal abnormal semantics.

been achieved in this field (Wang et al., 2022c,a,b; Kong et al., 2022; Qin and Song, 2022). Nonetheless, generating radiological reports remains a formidable undertaking. However, several aspects require further exploration, such as: 1) the presence of pronounced visual and textual data biases. Within medical data, normal images predominate the section containing descriptions of normal utterances in the reports. 2) Unlike natural images, radiological images frequently exhibit high similarity, posing challenges in extracting fine-grained visual details. The presence of noise in reports impacts network optimization. The descriptions of similar semantics may vary among different doctors. For instance, in Figure 1a, "cardiac" and "mediastinal" are used, whereas in Figure 1b, "cardiomediastinal" is employed. Moreover, it is plausible for the syntax of the same description to exhibit significant variation. Some normal descriptions may not be mentioned due to physicians' practices. For instance, in Figure 1b, "consolidation" and "effusion" are referenced, whereas in Figure 1a, they are disregarded. These sources of noise significantly impact network optimization, yet only limited research has been dedicated to addressing this issue.

---

[1]Our code is available at https://github.com/kzzjk/NADM

*Corresponding authors.

To address the aforementioned limitations, we propose a semantic decoupling network based on abnormal pattern memory for generating reports. In medical reports, the fundamental semantic structure comprises three components: anatomy, observation, and judgment. To mitigate the impact of noise in the reports, we optimize the visual extractor using abnormal semantics. This is based on the observation that abnormal semantics are relatively consistent, despite the occasional omission of normal descriptions and variations in similar semantic descriptions across reports from different doctors. We utilize RadGraph (Jain et al., 2021) to extract the semantics, focusing solely on the core semantic components while disregarding others. We employ a visual encoder based on higher-order feature interaction attention, enhancing the perception of fine-grained features through a bilinear pool (Kim et al., 2016), which has been shown to be effective in fine-grained classification (Lin et al., 2015). Furthermore, drawing inspiration from discrete variational autoencoder (Van Den Oord et al., 2017), we store the abnormal modalities of images and incorporate higher-order feature interaction processes to augment anomaly perception. During the report generation phase, we fuse the two modalities, semantic and visual, to jointly guide the report generation process. In the training phase, we employ convolution for image encoding and grid feature generation. Moreover, we utilize a variational self-encoder to capture anomaly modality memory and introduce two semantic branches atop it to predict both abnormal and normal semantics of the image during the visual attention phase. We argue that different words in the report possess varying levels of importance, with abnormal semantics being intuitively more significant than normal semantics. However, previous approaches (Jing et al., 2017; Yuan et al., 2019; Yin et al., 2019; Najdenkoska et al., 2021; Chen et al., 2021b; Wang et al., 2022c) tend to treat all report words equally. Therefore, we divide the reported words into four categories: normal/abnormal sentences and normal/abnormal semantics. Subsequently, we automatically learn distinct weighting parameters for optimization. In conclusion, our contributions are outlined as follows:

(1) We introduce a network for report generation that leverages anomaly semantic extraction. This approach focuses on optimizing the visual extraction network solely using anomaly semantics,

effectively mitigating the impact of noise and data bias present in reports.

(2) We develop a visual encoder based on anomaly pattern memory, which enhances anomaly perception by explicitly memorizing abnormal patterns and incorporating them during higher-order interaction in the visual processing phase.

(3) Our approach shows promising performance on MIMIC-CXR over multiple state-of-the-art methods.

## 2 Related Work

Image captioning involves generating relevant textual descriptions or topics for a given image. Early approaches utilized templates or retrieval-based methods for caption generation (Hossain et al., 2019). In recent years, encoder-decoder frameworks, primarily based on transformer architectures (Vaswani et al., 2017), have been widely employed to generate individual descriptive sentences from images (Cornia et al., 2020; Anderson et al., 2018; Chen et al., 2021a; Xu et al., 2021). Notably, anchor-frame-based methods for title generation have demonstrated accurate text description generation (Xu et al., 2021). However, a significant portion of the available data lacks annotations. To address this limitation, (Fang et al., 2022) proposed the use of semantic extraction to enrich information and improve caption generation. Furthermore, while most caption generation tasks focus on shorter descriptive discourse, existing approaches for very long utterances often overlook issues related to data bias (Melas-Kyriazi et al., 2018). The image captioning task has a more explicit objective, but the presence of complex semantic contexts, judgmental utterances, and noise problems in medical reports introduces additional challenges.

Medical report generation, which falls under the category of image captioning, predominantly adopts an encoder-decoder framework. Various attention mechanisms and hierarchical LSTM models have been proposed to generate radiology reports (Jing et al., 2017; Yuan et al., 2019; Yin et al., 2019). Another approach involves constructing graphs based on medical knowledge and utilizing graph convolutional neural networks to enhance feature extraction (Zhang et al., 2020). To address the issue of modal bias, (Najdenkoska et al., 2021) introduced a set of latent variables as topics to guide sentence generation by aligning image and language patterns in the latent space. Modal

alignment was further improved through the introduction of a cross-modal memory network by (Chen et al., 2021b), and later refined by (Wang et al., 2022a) with the proposition of a cross-modal prototype-driven network. Additionally, (Liu et al., 2021a) presented a framework that leverages both a priori and a posteriori data to enhance generative reporting. (Wang et al., 2022c) utilized semantic extraction to improve generation, they overlooked the problems of noise and data bias prevalent in the reports. In (Jing et al., 2019), the data bias problem is considered, where the noise of normal statements still affects the optimization of the model despite the separate generation of normal/abnormal statements.

## 3 The Proposed Method

The proposed architecture for the memory of anomaly patterns is the semantic decoupling network, consisting of three critical components: a visual extractor, a semantic extractor, and a decoder, where the visual extractor consists of an image encoder and a visual encoder. The overall structure of the network is illustrated in Figure 2. The visual extractor is responsible for converting images into detailed features with the goal of detecting anomalies and generating precise visual representations. The semantic extractor is divided into two parts: abnormal semantic extractor and normal semantic extraction. The purpose of abnormal semantic extractor is to identify semantics related to aberrations from the preceding visual representation and to capture normal semantics while considering the abnormal semantics. The decoder is responsible for processing the visual and semantic characteristics, and guides the report generation by deep fusion.

### 3.1 Image Encoder

To extract visual information from the images, we extracted features using the ResNet50 pre-trained by BioViL (Boecking et al., 2022). Given an image $I$, grid features are obtained by convolution $V_l = \mathrm{ResNet50(I)}$, and combined with global features $V_g$ with position encoding $E_{pos}$ to compose the visual information $V_c$.

$$V_c = Concat\left(V_g, V_l\right) + E_{pos} \qquad (1)$$

### 3.2 Abnormal Mode Memory

To prevent anomalies from being overwhelmed, we are inspired by variational autoencoder (Van

Den Oord et al., 2017), and introduce a discrete potential space called the memory codebook $\Omega$, where each entry in $\Omega$ corresponds to a potential embedding of an image pattern, and N denotes the total number of memorized patterns as a hyperparameter. In the case of an abnormal image, the features extracted by visual convolution search for a match within $\Omega$ to find the corresponding location of the memory pattern. While the nearest neighbor approach is a simple way to find a match, it poses the risk of pattern collapse (Chen et al., 2022). Thus, we adopt a dichotomous matching approach and apply the Hungarian algorithm (Kuhn, 1955) to resolve this issue. When memorizing, we look for a unique pattern in the memory space corresponding to each image encoding. The Hungarian algorithm is used to find a unique pattern match in the memory matrix by treating the image features with each pattern of the memory matrix as an ensemble matching problem. Specifically, we construct a bipartite graph using all local grid features $V_g$ and pattern embeddings in $\Omega$, ensuring that the sizes of $V_g$ and codebook $k$ are matched using $\emptyset$, and then determine the arrangement of the $k$ elements $\tau \in \mathfrak{S}_N$ with the lowest allocation cost.

$$\hat{\tau} = \arg\min_{\tau \in \mathfrak{S}_N} \sum_i^N \left\| V_g^i - \Omega_{\tau(i)} \right\|_2 \qquad (2)$$

### 3.3 Abnormal Pattern Enhancement Multi-Head Attention

The general self-attention approach only leverages the interaction of first-order features, which presents certain limitations. To achieve more detailed feature extraction, we adopt the bilinear attention module designed in accordance with (Pan et al., 2020). In the implementation, global features are queries $Q$, regional features are keys $K$, and values $V$. To incorporate abnormal patterns into higher-order interactions, we extend the key and value sets with mnemonics to encode and collect abnormal patterns from visual convolution. Our keys and values can be defined as $K = [K, \Omega]$ and $V = [V, \Omega]$, where $[,]$ denotes splicing and $\Omega$ is all abnormal embeddings of the memory. Then a low-rank bilinear pooling is performed to obtain the joint bilinear query-key $B_k$ and query-value $B_V$ by $\mathbf{B}_k = \sigma\left(\mathbf{W}_k \mathbf{K}\right) \odot \sigma\left(\mathbf{W}_q^k \mathbf{Q}\right)$ and $\mathbf{B}_V = \sigma\left(\mathbf{W}_V \mathbf{V}\right) \odot \sigma\left(\mathbf{W}_q^V \mathbf{Q}\right)$, $W_v, W_q^V, W_k$ and $W_q^K$ are learnable parameters. $\sigma$ denotes ReLU unit, and $\odot$ represents Hadamard Product. We then compute the attention on the space and channels by

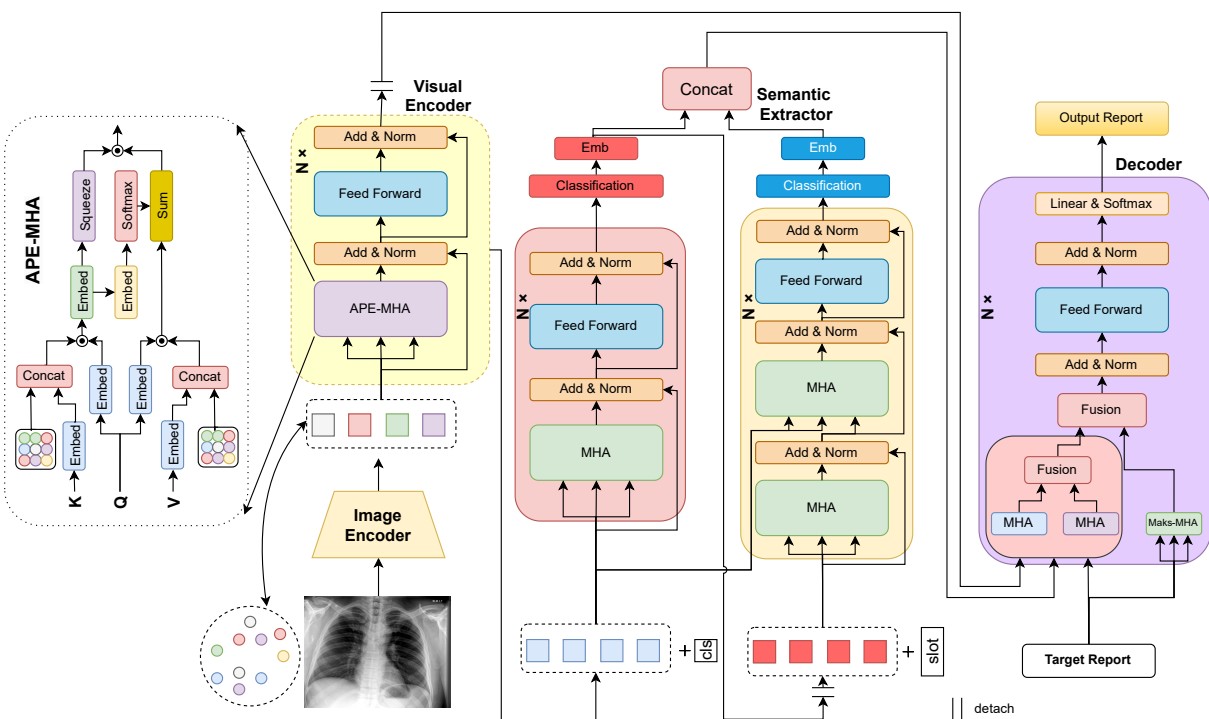

Figure 2: The overall structure of the model includes three modules: a visual extractor, a semantic extractor, and a decoder. The visual extractor module is a combination of an image encoder and a visual encoder. The images are encoded to obtain embedded features. Fine-grained information is obtained through high-level interactions by the visual encoder. The semantic extractor module decouples normal/abnormal semantics by the two branches. Finally, the report is generated through fusion decoding.

projecting each bilinear query key representation into the corresponding attention weights through two embedding layers, and then normalize using the softmax layer to introduce the spatial bilinear attention $\mathbf{B}'_k = \sigma\left(\mathbf{W}^k_B \mathbf{B_K}\right)$, then use another linear layer to map $B'_k$ from $D_c$ dimension to 1 dimension to obtain the spatial-wise attention weight $\alpha_s$. Meanwhile, we perform a squeeze-excitation operation (Hu et al., 2018) to generating channel attention $\beta_c = Sigmoid(W_c)\bar{B}$, $W_c$ are learnable parameters and $\bar{B}$ is an average pooling of $B'_k$. The basic layers that make up the visual encoder are as follows:

$$APE(V_c) = \beta_c \odot \alpha_s B_v \quad (3)$$

$$\mathbf{V}^i_f = AddLN\left(FFN\left(AddLN\left(APE\left(V_c\right)\right)\right)\right) \quad (4)$$

Where $V^i_f$, $V^i_f$ represents the i-th layer of visual feature extraction. FFN denotes a fully connected feed-forward network, and AddLN represents the composition of a residual connection and a normalization layer.

## 3.4 Semantic Extractor

To extract semantic information, we treat it as a multi-label classification process, and employ Rad-Graph (Jain et al., 2021) to extract pseudo-medical concepts as ground-truth for this task. The transformer attention block is used to process the intermediate features $V^m_f$ generated by the image encoder. To facilitate multi-label perception (MLP) network prediction, the [CLS] token is added to the image features, and its corresponding output represents the concept token. The vocabulary of concept tokens is the same as that used for the headings. Importantly, we predict concepts at the token level rather than the label level, to enable the multimodal decoding module to directly employ the top-K tokens for guidance reporting. The extraction process of anomaly semantics involves the following:

$$\text{Att}(Q, K, V) = \text{Softmax}\left(\frac{QK^T}{\sqrt{d}}\right)V \quad (5)$$

$$MHA = [Att_1, Att_2...]W^o \quad (6)$$

$$\mathbf{f}_m = MHA\left(V^m_f, V^m_f, V^m_f\right) \quad (7)$$

$$\mathbf{S}_a = AddLN\left(FFN\left(AddLN\left(f_m\right)\right)\right) \quad (8)$$

$$\mathbf{S}_a^K = Emb(MLP(S_a^1)) \tag{9}$$

Where $S_a$ is the generated intermediate semantic prediction, $S_a^1$ denotes the output corresponding to the classification flag [CLS], MLP represents the final classification layer, Emb represents the embedding layer, Att represents the attention layer, and $S_a^K$ represents the top-k abnormal semantic concepts encoded at the end.

After obtaining the abnormal semantics, we consider the acquisition of normal semantics as a conditional probability problem under the condition of abnormal semantics. Due to the complexity of the normal semantics, it does not serve as a basis for optimal visual extractor, so its update is only related to the normal semantic extractor. We add a [slot],representing the prediction of the normal semantics. The extractor process is as follows:

$$\mathbf{f}_n = AddLN(MHA\left(S_a^K, S_a^K, S_a^K\right)) \tag{10}$$

$$\mathbf{f}_n^{'} = MHA(f_n, V_f^m, V_f^m) \tag{11}$$

$$\mathbf{S}_n = AddLN\left(FFN\left(AddLN\left(f_n^{'}\right)\right)\right) \tag{12}$$

$$\mathbf{S}_n^K = Emb(MLP(S_n^1)) \tag{13}$$

$$\mathbf{S}_f = Concat(S_a^K, S_n^K) \tag{14}$$

Where $S_n$ is the generated intermediate semantic prediction, $S_n^1$ denotes the output corresponding to the classification flag [slot] , $S_n^K$ represents the top-k normal semantic concepts encoded at the end and $S_f$ represents the final semantic output.

### 3.5 Decoder

We adopt the decoder structure introduced in (Li et al., 2022), which leverages the rich visual tokens $V_f$ obtained from the visual encoder and the semantic tokens $S_f$ extracted from the semantic extractor. The decoder combines both visual and semantic information to provide guidance for generating accurate and coherent sentences. Formally, let $R = \{r_0, r_1, ..., r_t\}$ denote report of t words generated. The decoder takes the report R as input and learns to predict the next word automatically and regressively, conditional on the visual $V_f$ and semantic tokens $S_f$. We implemented the decoder as an Nd-stacked transformer block. It consists of a masked multiheaded attention layer Mask-MHA for modeling the overall textual context $h_t^{'}$ of the previously generated word $h_t$, and two crossed multiheaded attention layers that cross the visual and semantic tokens to trigger the generation of the cue

$h_t^v$, respectively. subsequently, the previous contextual, semantic, and visual information is fused and encoded using Sigmoid to obtain the gating $g$.Taking the state at time t as an example, the equation is as follows:

$$\mathbf{h}_t^{'} = Mask\text{-}MHA(h_t, h_t, h_t) \tag{15}$$

$$\mathbf{h}_t^v = MHA(h_t, V_f, V_f) + MHA(h_t, S_f, S_f) \tag{16}$$

$$\mathbf{g} = Sigmoid(W_g[h_t^v, h_t^{'}]) \tag{17}$$

$$\mathbf{h}_{t+1} = AddLN(gh_t^{'} + (1 - g)h_t^v) \tag{18}$$

### 3.6 Objective Function

**Memory loss:** the anomaly memory codebook serves the purpose of storing various anomaly information and interacting with image information. It is important to note that the learning process of the codebook should not affect the image coding. We want to use orthogonality to make the codebook learn as much information as possible about different modes. We use Sg to represent the gradient cutoff and E to denote the unit matrix. The loss function is defined as follows:

$$\mathcal{L}_m = \left\| sg\left[V_c^i\right] - q\left(V_c^i\right)\right\|_2^2 + \left\|\mathbf{\Omega}^\top\mathbf{\Omega} - E\right\|^2 \tag{19}$$

**Semantic loss:** in the semantic extraction task, our goal is to extract normal and abnormal semantics, which entails a multi-label classification process. However, the semantic distribution in medical reports is often heavily imbalanced, which can pose a challenge for standard multi-label classification approaches. To address this issue, we employ an asymmetric loss (Fang et al., 2022; Ridnik et al., 2021; Liu et al., 2021b; Ridnik et al., 2021), which has shown good performance in handling unbalanced problems.

$$\mathcal{L}_s = asym\left(\tilde{P}_a, \mathbf{y}_a\right) + asym\left(\tilde{P}_n, \mathbf{y}_n\right) \tag{20}$$

$\tilde{P}_a, \tilde{P}_n$ denotes the predicted probability distribution corresponding to the abnormal/normal semantics, respectively, and $y_a, y_n$ denotes its corresponding label.

**Reports loss:** For the generated reports, we use minimizing the negative log-likelihood of the given image features, semantic features to train the model parameters. Although our positive anomaly semantic decoupling avoids the effect of noise on feature extraction, the effect of noise is still unavoidable in

Table 1: The performances of our model compared with baselines on MIMIC-CXR dataset. The best results are highlighted in bold.

| Dataset | Model | BLEU-1 | BLEU-2 | BLEU-3 | BLEU-4 | METEOR | ROUGE-L |
|---------|-------|--------|--------|--------|--------|--------|---------|
| MIMIC-CXR | S&T | 0.299 | 0.184 | 0.121 | 0.084 | 0.124 | 0.263 |
| | AdaAtt | 0.299 | 0.185 | 0.124 | 0.088 | 0.118 | 0.266 |
| | TopDown | 0.317 | 0.195 | 0.130 | 0.092 | 0.128 | 0.267 |
| | R2Gen | 0.353 | 0.218 | 0.145 | 0.103 | 0.142 | 0.277 |
| | PPKED | 0.360 | 0.224 | 0.149 | 0.106 | 0.149 | 0.284 |
| | M2TR | 0.378 | 0.232 | 0.154 | 0.107 | / | 0.272 |
| | R2GenCMN | 0.353 | 0.218 | 0.148 | 0.106 | 0.142 | 0.278 |
| | XProNet | 0.344 | 0.215 | 0.146 | 0.105 | 0.138 | 0.279 |
| | GSKET | 0.363 | 0.228 | 0.156 | 0.115 | / | 0.284 |
| | R2GenRL | 0.381 | 0.232 | 0.155 | 0.109 | 0.151 | 0.287 |
| | MSAT | 0.373 | 0.235 | 0.162 | 0.120 | 0.143 | 0.282 |
| | **Ours** | **0.402** | **0.258** | **0.179** | **0.130** | **0.155** | **0.289** |

the decoding phase of report generation. Thus, measuring the importance of different words with different weights to mitigate the effect of noisy normal descriptions makes the network focus more on critical information. We use RadGraph to classify the vocabulary in the report into normal/abnormal sentences, and normal/abnormal semantics. Inspired by the multitasking approach (Kendall et al., 2018; Liebel and Körner, 2018), the importance of words is measured using uncertainty. Following (Liebel and Körner, 2018), the four weights are learned automatically.

$$\mathcal{L}_r^{'} = \sum_{i=1}^{T} \log P_\theta \left( \mathbf{R}_t \mid \mathbf{R}_{<t}, \mathbf{V_f}, \mathbf{S_f} \right) \quad (21)$$

$$\mathcal{L}_r = \sum_{m=0}^{4} \frac{1}{\sigma_m^2} L_r^m + log(1 + \sigma_m) \quad (22)$$

$R_t$ denotes the report information at time t, $V_f$ denotes the image information, $S_f$ denotes the semantic information, $\sigma_m$ denotes the different uncertainty parameters. $L_r^m$ denotes category m reports loss. The final losses are as follows:

$$\mathcal{L} = \mathcal{L}_m + \mathcal{L}_s + \mathcal{L}_r \quad (23)$$

# 4 Experiment Settings

## 4.1 Datasets

We conducted numerical experiments on MIMIC-CXR[2] (Johnson et al., 2019). MIMIC-CXR is the largest radiology dataset to date, including 473,057

[2]https://physionet.org/content/mimic-cxr/2.0.0/

Table 2: The results of clinical efficacy (CE) metrics on the MIMIC-CXR dataset. The best results are highlighted in bold.

| Model | Precision | Recall | F1-Score |
|-------|-----------|--------|----------|
| S&T | 0.249 | 0.203 | 0.204 |
| AdaAtt | 0.268 | 0.186 | 0.181 |
| TopDown | 0.320 | 0.231 | 0.238 |
| R2Gen | 0.333 | 0.273 | 0.276 |
| R2GenCMN | 0.334 | 0.275 | 0.278 |
| R2GenRL | 0.342 | 0.294 | 0.292 |
| **Ours** | **0.417** | **0.413** | **0.415** |

chest x-ray images. In our experiments, for a fair comparison, we used the official segmentation of MIMIC-CXR after the work (Chen et al., 2020) and excluded all sample reports that did not contain a description of medical observations. We only focus on the finding part of the medical report.

## 4.2 Evaluation Metrics

We used the widely used BLEU (Papineni et al., 2002), METEOR (Banerjee and Lavie, 2005), and ROUGE-L (Lin, 2004), computed by the standard evaluation toolkit [3]. In particular, BLEU and METEOR are proposed for machine translation evaluation. ROUGE-L was designed to measure the quality of the summary. For report generation, the predictive accuracy of the disease should also be considered. Therefore, we used clinical efficiency (CE) metrics to express the performance of our model. We used CheXpert (Irvin et al., 2019)[4] to

[3]https://github.com/tylin/coco-caption
[4]https://github.com/MIT-LCP/mimic-cxr/tree/master/txt/chexpert

Table 3: The performances of our model compared with the model without different proposed modules on MIMIC-CXR dataset. The w/o is the abbreviation of without. The best results are highlighted in bold.

| MIMIC-CXR | BLEU-1 | BLEU-2 | BLEU-3 | BLEU-4 | METEOR | ROUGE-L |
|-----------|--------|--------|--------|--------|--------|---------|
| Ours | 0.402 | **0.258** | **0.179** | **0.130** | 0.155 | **0.289** |
| w/o Bio | **0.405** | 0.255 | 0.171 | 0.121 | 0.156 | 0.286 |
| w/o SE | 0.403 | 0.254 | 0.171 | 0.120 | 0.156 | 0.284 |
| w/o APE | 0.396 | 0.250 | 0.169 | 0.119 | **0.161** | 0.284 |
| w/o WE | 0.398 | 0.251 | 0.172 | 0.124 | 0.158 | **0.289** |

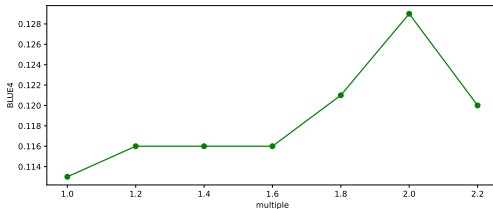

Figure 3: The BLEU-4 scores and versus memory size when the model is trained and tested on the MIMIC-CXR dataset. The horizontal coordinate indicates the size of the multiplier of the base capacity 49.

extract disease labels from real reports and model predictions to calculate precision, recall, and F1 scores.

**An important note:** the implementation details, dataset splits, preprocessing steps, and additional experiments are described in the supplementary materials.

# 5 Results and Analysis

## 5.1 Comparison with Previous Studies

To further validate the effectiveness of our method, we compare our proposed method with conventional image captioning works, e.g. **S&T** (Vinyals et al., 2015), **AdaAtt** (Lu et al., 2017), **Top-Down** (Anderson et al., 2018), and the ones proposed for the medical domain, e.g. **R2Gen** (Chen et al., 2020), **PPKED** (Liu et al., 2021a), **M2TR** (Nooralahzadeh et al., 2021), **R2GenCMN** (Chen et al., 2021b), **XProNet** (Wang et al., 2022a), **GS-KET** (Yang et al., 2022), **R2GenRL** (Qin and Song, 2022) and **MSAT**(Wang et al., 2022c).The results on S&T (Vinyals et al., 2015), AdaAtt (Lu et al., 2017), TopDown (Anderson et al., 2018) from (Chen et al., 2020), and the rest of the results were cited from the original paper. Table 1 shows the NLG metrics and Table 2 shows the CE metrics. As can be seen, thanks to our good framework of positive anomaly separation and excellent feature extraction ability. We achieve a large improvement

in both metrics and obtain SOTA results. This proves the superiority of our method.

## 5.2 Ablation Study

In this section, we performed ablation experiments on the MIMIC dataset to investigate the contribution of each component of our method. Table 3 shows the experimental results. Bio denotes the initialization of the ResNet network weights using BioViL (Boecking et al., 2022). DE denotes the truncated reports loss on the visually extracted gradients. SE denotes the semantic extractor structure with normal and abnormal separation. APE denotes the abnormal memory extraction structure. WE denotes the report loss weight adjustment.We investigated the following variants:

**w/o Bio** indicates that the model for ResNet50 randomly initializes the weight parameters and leaves the rest of the structure unchanged. **w/o SE** indicates that instead of using decoupled extraction semantics, a single semantic extractor branch is taken and normal and abnormal semantics are predicted simultaneously. **w/o APE** indicates that instead of using the structure of APE-MHA, the visual encoder is replaced with the structure of self-attention MHA. **w/o WE** indicates that instead of weight the report loss, but treat all words in the report equally.

**Contribution of each component.** We conducted an analysis to evaluate the importance of different components in our proposed method. We found that all key components play a critical role in achieving high performance. Removing any of these components results in significant performance degradation. Specifically, if we do not initialize our visual extractor with BioVil, we observe a reduction in performance. This underscores the importance of BioVil in providing prior information by aligning images with reports, thereby reducing the modal differences. This also suggests that some alignment aspects could potentially be

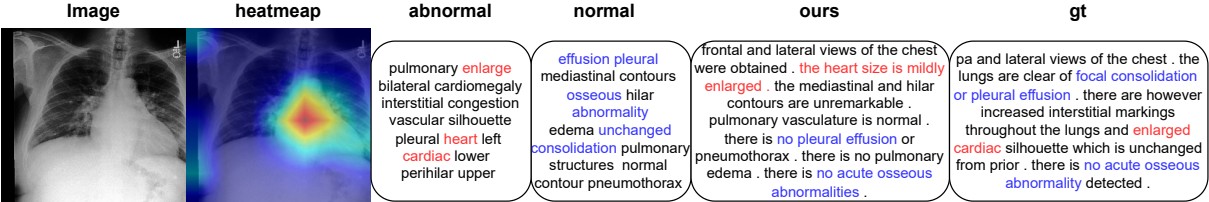

| Image | heatmeap | abnormal | normal | ours | gt |
|---|---|---|---|---|---|
| | | pulmonary enlarge bilateral cardiomegaly interstitial congestion vascular silhouette pleural heart left cardiac lower perihilar upper | effusion pleural mediastinal contours osseous hilar abnormality edema unchanged consolidation pulmonary structures normal contour pneumothorax | frontal and lateral views of the chest were obtained . the heart size is mildly enlarged . the mediastinal and hilar contours are unremarkable . pulmonary vasculature is normal . there is no pleural effusion or pneumothorax . there is no pulmonary edema . there is no acute osseous abnormalities . | pa and lateral views of the chest . the lungs are clear of focal consolidation or pleural effusion . there are however increased interstitial markings throughout the lungs and enlarged cardiac silhouette which is unchanged from prior . there is no acute osseous abnormality detected . |

Figure 4: Model visualization, where the heatmap shows the anomalies of interest to the model;red text indicates the abnormal semantics related to the report; blue text indicates the normal semantics related to the report.

integrated with our method to further boost performance. Additionally, we found that the decoupling structure and APE-MHA are crucial components, and their absence results in the largest performance degradation. This highlights the significance of decoupled learning, which mitigates the negative impact of the uncertainty noise contained in normal semantics on predicting abnormal semantics. Furthermore, our proposed abnormal modal memory structure enhances the perception of fine-grained images and improves model performance. We also observed that WE enhances the report generation process by optimizing the grouping of words in the report and assigning different weights to them. This allows the decoder to focus more on the key semantics, resulting in better quality reports. Overall, our analysis demonstrates the effectiveness and importance of each component in our proposed method.

**Impact of memory size.** To analyze the effect of memory capacity on the model during anomaly modality extraction, we trained the model using different memory capacities. Since the memory modality is matched one-to-one with the local features of the image, using a multiple of the image information size of 49 we conducted the experiments and the results are shown in Figure 3. It can be seen that as the memory capacity increases, the model performance rises. This is due to the fact that a larger capacity allows more image information to be stored and may capture more anomalies, which can be used as a prior for subsequent bilinear extraction. However, when greater than a certain threshold, the memory of abnormal modes is sufficient, and more capacity instead memorizes unimportant secondary information introducing noise and causing negative effects.

### 5.3 Qualitative analysis

To evaluate the performance of our model, we performed a qualitative analysis on the MIMIC-CXR dataset. As shown in the Figure 4, we visualized the attention graph produced by the model for the images, as well as for the top 15 predicted semantics. Abnormal semantics related to the report were highlighted in red, while normal semantics related to the report were highlighted in blue. Our analysis revealed that the model pays close attention to abnormal regions in the images, and successfully predicts abnormal phenomena such as heart enlargement. In the prediction of normal semantics, the model gives a rich set of normal semantic candidates, containing all normal mentions in real reports, such as consolidation, pleural effusion, and osseous abnormality. This shows that our model can produce relatively accurate and fluent reports. These results suggest that our model is capable of generating relatively accurate and fluent reports.In addition, we visualize the final learned report weights such that $\lambda = \frac{1}{\sigma^2}$, then $\lambda$ corresponding to abnormal normal semantics, abnormal normal sentences is [1.502,1.196,0.993, 0.691]. From the weights, we can see that the abnormal cases have higher weights and the semantics have higher weights than the normal sentences. This indicates that the main body of the report is the anomaly semantics, which contains more information. It is more important to pay attention to the anomaly semantics when the report is generated.

## 6 Conclusions

We propose a new framework for extracting normal and abnormal semantics separately. By decoupling the semantics, the noise that may exist in the normal semantics is avoided. And we propose the structure of anomaly modality enhancement to enhance the extraction of abnormal features. We classify the reported words into different parts to automatically learn the weights to measure the importance. This eventually makes better semantic and visual features jointly know to generate process accurate reports.

## Limitations

While our approach achieves good results, it is not without limitations. The main limitation lies in treating semantic extraction as a multi-label classification process. In practice, medical reports contain many labeling concepts, including some rare diseases that are seldom mentioned and difficult for the network to identify. And some specific descriptions such as "acute obliquely oriented lucency through the right 12th posterior rib" cannot be generated.Additionally, while noise in the report does not affect visual feature extraction, optimizing the report using different weights may only partially alleviate the effects of noise on the decoder, without addressing the root cause.In the future, it may be necessary to start at the data level and standardize reporting up front. Reduce the interference of uncertainty. For rare diseases, additional medical knowledge may need to be introduced into the model. Improve identification of rare diseases through external knowledge enhancement. So at this stage, it can still only play a supporting role in the medical system.

## Ethics Statement

The MIMIC-CXR datasets employed in our study underwent a meticulous de-identification process. Our utilization of the MIMIC-CXR dataset aligns with the PhysioNet Health Data license 1.5.0 7. Importantly, the distributed MIMIC-CXR dataset underwent thorough processing to remove all instances of protected health information (PHI). This approach underscores our commitment to adhering to ethical standards.

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

## A  Appendix

### A.1  Dataset and pre-processing

We used the MIMIC-CXR dataset consisting of 377,110 chest x-ray images. To facilitate comparison, we followed the preprocessing methods of other previous work (Chen et al., 2020). We only focus on the finding part of the medical report, exclude invalid data in the data, and use lowercase for all reports. Words that appear less than 10 times are ignored and replaced with <unk>. Only the first 100 words are retained for each report. The final number of training/validation/testing sets obtained based on official data division: 269235/2112/3851. In the training process, we scaled the image data to a uniform size of (256,256) and used the following data enhancements: random image cropping to (224,224) and affine transformation, translation up to ±2% of image height/width and rotation up to ±10°.

Table 4: The impact of different semantics and extraction methods on the proposed model on the MIMIC-CXR dataset.

| Setting | SE | Abnormal | Normal | BLEU-1 | BLEU-2 | BLEU-3 | BLEU-4 | METEOR | ROUGE-L |
|---------|-----|----------|--------|--------|--------|--------|--------|--------|---------|
| a | | ✓ | | 0.396 | 0.248 | 0.167 | 0.120 | 0.150 | 0.288 |
| b | | ✓ | ✓ | 0.403 | 0.254 | 0.171 | 0.120 | 0.156 | 0.284 |
| c | ✓ | ✓ | ✓ | 0.402 | 0.258 | 0.179 | 0.130 | 0.155 | 0.289 |

Table 5: The proposed model's performance compared to those different memory modules on the MIMIC-CXR dataset.

| Setting | BLEU-1 | BLEU-2 | BLEU-3 | BLEU-4 | METEOR | ROUGE-L |
|---------|--------|--------|--------|--------|--------|---------|
| a | 0.354 | 0.222 | 0.152 | 0.109 | 0.132 | 0.267 |
| b | 0.402 | 0.258 | 0.179 | 0.130 | 0.155 | 0.289 |

To enable semantic tagging, we utilize the Rad-Graph (Jain et al., 2021) method for extracting clinical entities and their relationships from radiology reports. Specifically, we focus on extracting anatomical entities and observation entities from the reports. To perform normal/abnormal semantic classification, we employ a keyword detection method. In particular, we define a set of normal keywords based on the reference (Yu et al., 2022). These extracted semantics are then used as multi-label classification labels for the semantic extraction task. Purely keyword-based detection sometimes produces misclassifications due to keywords. For example, 'enlarged heart size is stable since' may be misclassified as normal. To address these challenges, we have leveraged ChatGPT to enhance our categorization process. Specifically, the formidable semantic understanding capabilities of ChatGPT assist us in discerning between normal and abnormal cases. We use the GPT-3.5-turbo model for a two-stage assessment. In the first stage, we determine whether the sentence is normal or abnormal. In the second stage, for sentences identified as abnormal, we use the semantic keywords extracted by RadGraph to assess whether the specific semantics are normal or abnormal. In the first stage, prompt: "You are a specialized radiologist. Evaluate the following medical descriptions. Note: any deviation from the imaging presentation of a normal person is considered abnormal. No need to explain; answer directly: normal or abnormal. Description: XX." In the second stage, prompt: "You are a specialized radiologist. Determine whether the description of a given keyword is abnormal in a medical description. Note: any deviation from the imaging presentation of a normal person is considered abnormal. No need to explain, answer directly:

normal or abnormal. Description: XX. Keyword: XX." It is important to note that our semantics share the same word list as the original report, and do not require any additional tokenization. Furthermore, the sentences where normal semantics are found are categorized as normal sentences, while those containing abnormal semantics are labeled as abnormal sentences. Overall, we divide the words in the report into four parts: normal/abnormal sentences, and normal/abnormal semantics.

## A.2 Implementation Details

To ensure consistency with the experimental setup of previous work, we used one image as input for MIMIC-CXR. The encoder and decoder modules in our framework consist of three basic layers. We set the 8 attention heads, 512 dimensions for hidden states and initialize it randomly respectively. During the report generation process, we set the beam size to 3. The codebook size N is 98, while the value of K used in the semantic generation process is 25. Semantic branch starts from the first layer APE-MHA. To optimize the model, we employ the AdamW optimizer, with a model learning rate set to $5 \times 10^{-6}$. We implement our model using Pytorch and X-modaler (Li et al., 2021). We used 2080ti at MIMIC-CXR for 30 epoch of training for a total of 26 GPU hours.

## A.3 Additional Experiments

### A.3.1 Semantic extraction method

We evaluated the effectiveness of our proposed method for semantic extraction, and the results are summarized in Table 4. In the table, SE represents the decoupled extraction of semantics, where Abnormal and Normal indicate abnormal and normal semantics, respectively. Specifically, a denotes the

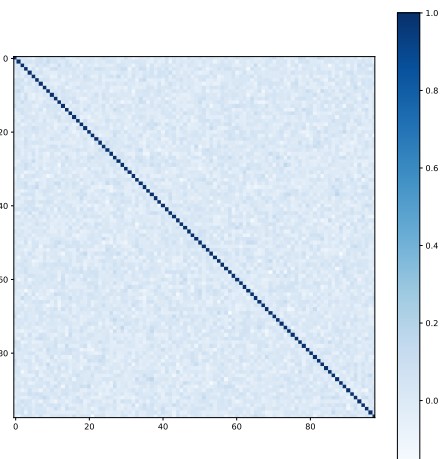

Figure 5: Cosine similarity scores of all patterns memorized in the codebook Ω.

extraction of abnormal semantics only, b denotes the extraction of both normal and abnormal semantics but without decoupling, and c represents our proposed decoupling method. Comparing a and b, we observe that both types of semantics have a positive impact on the final report generation. However, when comparing b and c, we find that the decoupled and separated extraction approach is more effective, with a 0.9% improvement in BLUE4. This underscores the presence of noise in normal semantics, which can adversely affect the visual extractor and compromise reporting accuracy. On the other hand, our proposed decoupling method avoids the effects of noise and achieves better reporting results.

### A.3.2 Memory method

We compared our proposed memory approach with the memory modeling approach proposed in method (Cornia et al., 2020) to highlight the role of abnormal memory. For the experiments, we kept parameters such as memory capacity constant. The results show that the memory approach in method (Cornia et al., 2020) did not perform well. We believe that this is due to data bias, as method (Cornia et al., 2020) introduces a memory matrix in the self-attentive phase, which enables it to automatically memorize features in the training data and form prior knowledge. However, since data bias may have caused it to memorize more repetitions of normal cases, it achieved suboptimal results.

In contrast, our approach provides more useful information by explicitly memorizing abnormali-

ties. Specifically, we memorize only the images where anomalies exist, and the model is encouraged to memorize different features by leveraging one-to-one matching and mutual exclusion loss. Figure 5 shows the similarity matrix of the memorized codebooks in our method, and it can be observed that the patterns memorized in the codebooks only have a high similarity to themselves. This illustrates the diversity of our method's memory. Overall, our abnormal memory approach yields better results than method A, which underscores its effectiveness in addressing data bias and improving the performance of medical report generation models.

### A.3.3 Noise impact experiment

To explore the effect of noise in report and normal semantics, we designed additional experiments with Normal_De denoting the gradient of truncated normal semantic branches on visual extraction and Report_De denoting the gradient of truncated report decoding on visual extraction. a denotes that report noise with normal semantic noise can affect visual extraction, b denotes that report noise only affects, and c denotes that normal semantic noise only influence. d indicates our final model, i.e., truncated with all noise gradients. The results in Table 6 show that both noises have a negative impact and cause a decrease in the metrics. This also shows that our framework of normal anomaly semantic decoupling can avoid the effect of this complex noise to some extent.

### A.3.4 More detailed evaluation

to comprehensively demonstrate the performance of our model, we employed the RadGraph-F1 (Yu et al., 2023) test model and provided a detailed breakdown of disease-specific F1 scores and proportions. Table 9 represents the RadGraph-F1 scores, where R2Gen (Chen et al., 2020) and R2GenCMN (Chen et al., 2021b) are obtained by calculations using the officially provided model weights and X-REM (Jeong et al., 2023) are from the original paper. Our metrics achieve the highest scores. Table 8 represents the F1 scores for the 14 diseases. The weight of this disease occupying all the MIMIC-CXR data is labeled after each category. R2Gen (Chen et al., 2020) and R2GenCMN (Chen et al., 2021b) obtained by calculations using the officially provided model weights. It can be seen that the best scores are achieved in most categories. And the percentage of anomalies is small in terms of category share. Detection failures can be due

Table 6: Effect of different noise on the proposed model on the MIMIC-CXR dataset.

| Setting | Normal_De | Report_De | BLEU-1 | BLEU-2 | BLEU-3 | BLEU-4 | METEOR | ROUGE-L |
|---------|-----------|-----------|--------|--------|--------|--------|--------|---------|
| a |  |  | 0.397 | 0.251 | 0.172 | 0.124 | 0.151 | 0.289 |
| b | ✓ |  | 0.394 | 0.253 | 0.176 | 0.128 | 0.151 | 0.290 |
| c |  | ✓ | 0.391 | 0.250 | 0.173 | 0.125 | 0.150 | 0.289 |
| d | ✓ | ✓ | 0.402 | 0.258 | 0.179 | 0.130 | 0.155 | 0.289 |

Table 7: The performances of our model compared with baselines on IU-Xray dataset. The best results are highlighted in bold.

| Dataset | Model | BLEU-1 | BLEU-2 | BLEU-3 | BLEU-4 | METEOR | ROUGE-L |
|---------|-------|--------|--------|--------|--------|--------|---------|
| IU-Xray | S&T | 0.216 | 0.124 | 0.087 | 0.066 | / | 0.306 |
|  | AdaAtt | 0.220 | 0.127 | 0.089 | 0.068 | / | 0.308 |
|  | R2Gen | 0.470 | 0.304 | 0.219 | 0.165 | 0.187 | 0.371 |
|  | PPKED | 0.483 | 0.315 | 0.224 | 0.168 | / | 0.376 |
|  | M2TR | 0.486 | 0.317 | 0.232 | 0.173 | 0.192 | **0.390** |
|  | R2GenCMN | 0.475 | 0.309 | 0.222 | 0.170 | 0.191 | 0.375 |
|  | GSKET | 0.496 | 0.327 | 0.238 | 0.178 | / | 0.381 |
|  | R2GenRL | 0.494 | 0.321 | 0.235 | 0.181 | 0.201 | 0.384 |
|  | MSAT | 0.481 | 0.316 | 0.226 | 0.171 | 0.190 | 0.372 |
|  | Ours | **0.506** | **0.344** | **0.256** | **0.198** | **0.211** | **0.390** |

Table 8: The detailed results across all CheXpert categories for previous report generation models and our proposed model based on MIMIC-CXR.

| Class(%) | R2Gen | R2GenCMN | Ours |
|----------|-------|----------|------|
| No Finding (33.119%) | 0.512 | 0.513 | 0.471 |
| Enlarged Cardiomediastinum (7.266%) | 0.360 | 0.335 | 0.344 |
| Cardiomegaly (22.336%) | 0.377 | 0.405 | 0.406 |
| Lung Lesion (3.259%) | 0.329 | 0.327 | 0.322 |
| Lung Opacity (24.297%) | 0.350 | 0.350 | 0.364 |
| Edema (17.641%) | 0.385 | 0.428 | 0.429 |
| Consolidation (6.632%) | 0.335 | 0.354 | 0.370 |
| Pneumonia (15.295%) | 0.347 | 0.350 | 0.355 |
| Atelectasis (24.639%) | 0.369 | 0.398 | 0.341 |
| Pneumothorax (5.044%) | 0.338 | 0.339 | 0.348 |
| Pleural Effusion (26.386%) | 0.383 | 0.463 | 0.464 |
| Pleural Other (1.218%) | 0.326 | 0.331 | 0.331 |
| Fracture (2.171%) | 0.322 | 0.322 | 0.328 |
| Support Devices (29.318%) | 0.506 | 0.522 | 0.467 |

Table 9: Comparison of our model to previous report generation models on MIMIC-CXR.

| Model | RadGraph-F1 |
|-------|-------------|
| R2Gen | 0.165 |
| R2GenCMN | 0.182 |
| X-REM | 0.181 |
| Ours | 0.209 |

to different reasons, such as too little training data (e.g., "fracture", "Lesion") or too difficult to learn (e.g., "pneumothorax", which is is also difficult for clinicians to determine).

### A.3.5 IU-Xray experiment

The Indiana University chest x-ray collection (IU-Xray[5]) (Demner-Fushman et al., 2016) is a public radiology examination dataset and a common dataset used in medical report generation tasks.

---
[5]https://openi.nlm.nih.gov/

The dataset includes 7,470 x-ray images and the corresponding 3,955 radiology reports. However, the IU-Xray dataset does not have a standard dataset segmentation, resulting in some of the previous methods not performing comparably in terms of metrics. To facilitate comparison with previous work, we used the same preprocessing and segmentation as in (Chen et al., 2020). We also compare only the methods that use the same segmentation. The training/test/val setting for the entire dataset was 7:1:2.Our data processing steps were similar to those of MIMIC-CXR. Due to the small number of IU-Xray data, we excluded words with less than 3 occurrences and focused only on the first 60 words.

we compare our proposed method with conventional image captioning works, e.g. **S&T** (Vinyals et al., 2015), **AdaAtt** (Lu et al., 2017), and the ones proposed for the medical domain, e.g. **R2Gen** (Chen et al., 2020), **PPKED** (Liu et al., 2021a), **M2TR** (Nooralahzadeh et al., 2021), **R2GenCMN** (Chen et al., 2021b) **GSKET** (Yang et al., 2022), **R2GenRL** (Qin and Song, 2022) and **MSAT**(Wang et al., 2022c). The results on S&T (Vinyals et al., 2015), AdaAtt (Lu et al., 2017) from (Chen et al., 2020), and the rest of the results were cited from the original paper.As we can see from the results in Table 7, our method can still achieve good results on the IU-Xray dataset, which indicates the generality of our method.

### A.4 Hallucination samples

In order to more clearly exemplify the error and hallucination samples, we have listed a detailed comparison in Figure 6. It can be seen that our report (a) observes atelectasis but omits fractures. This may be mainly due to the fact that the fractures category is underrepresented in the data. Secondly we are not able to report specific descriptions like l1 and t12. Although we report degenerative changes, they are not specific to the right shoulder. For some of the normal descriptions of (b), we don't match well with the real report, we don't report no acute bony abnormalities. This has to do with the normal noise in the report, which is mentioned in some doctors' reports and not in others. In (c) we report patchy opacities, due to lung atelectasis, but in reality it is basilar lung opacities. and we report pleural effusion on the left side, but in the actual report it is present on the left side as well as the right side. It is possible that the model discriminated the pleural effusion but was inadequate for orientation. This

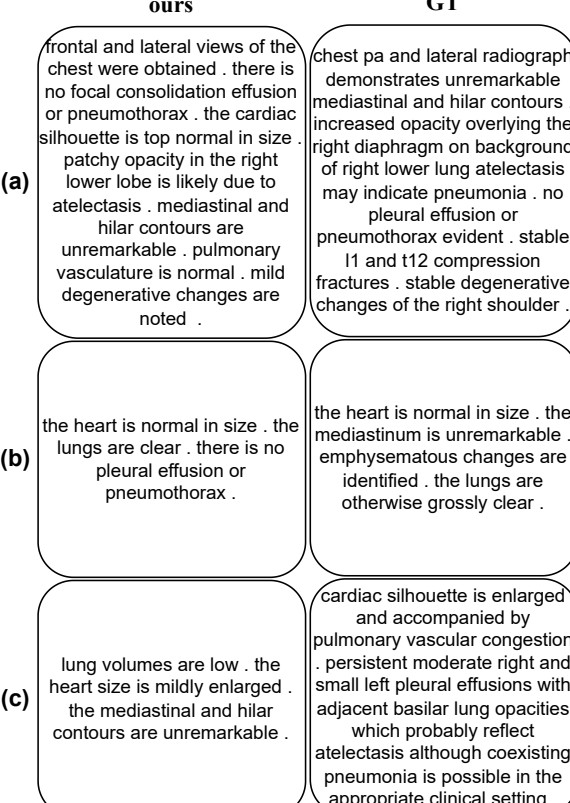

Figure 6: Example of a generated report compared to a real report.

may be due to the lack of more detailed labeling in the report.

### A.5 More Example Visualizations

| image | ours | GT |
|---|---|---|

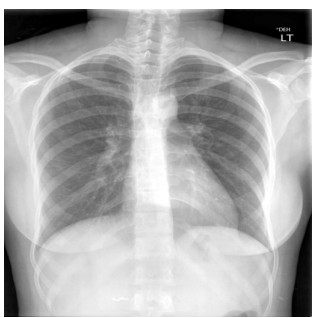

heart size and mediastinal contours are within normal limits . the lungs are clear . there is no pleural effusion or pneumothorax .

the heart size and mediastinal contours appear within normal limits . no focal airspace consolidation pleural effusion or pneumothorax . no acute bony abnormalities .

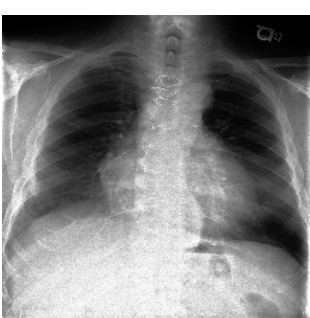

frontal and lateral views of the chest were obtained . the lungs are clear without focal consolidation effusion or pneumothorax . the cardiac silhouette is mildly enlarged . mediastinal and hilar contours are within normal limits . there is no pulmonary edema . there is no pleural effusion or pneumothorax . there is mild degenerative changes in the thoracic spine .

as compared to the previous radiograph there is no relevant change . no new parenchymal opacity . unchanged moderate cardiomegaly and unchanged position and course of the sternal wires and clips after cabg . the pre-existing platelike atelectasis in the left mid lung has resolved . unchanged area of mild right lateral pleural thickening . no pulmonary edema . no pleural effusions . no lung nodules or masses .

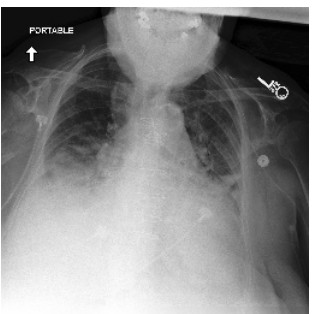

lung volumes are low . the heart size is mildly enlarged . the mediastinal and hilar contours are unremarkable . there is mild pulmonary vascular congestion . patchy opacities in the right lower lung base likely due to atelectasis . small bilateral pleural effusions are seen on the left . there is no pneumothorax . there is no pulmonary edema .there is no acute osseous abnormalities .

cardiac silhouette is enlarged and accompanied by pulmonary vascular congestion . persistent moderate right and small left pleural effusions with adjacent basilar lung opacities which probably reflect atelectasis although coexisting pneumonia is possible in the appropriate clinical setting .

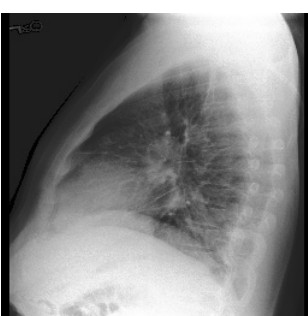

the heart is normal in size . the lungs are clear . there is no pleural effusion or pneumothorax .

the heart is normal in size . the mediastinum is unremarkable . emphysematous changes are identified . the lungs are otherwise grossly clear .

Figure 7: Visualization of prediction results, where GT is the abbreviation of the ground truth, text in red indicates the reported keywords and ground truth keywords generated by our method.