# OpenReview forum: "Normal-Abnormal Decoupling Memory for Medical Report Generation"
_EMNLP/2023/Conference — EMNLP 2023 Findings_

### Official Review · Reviewer_VhyG · 2023-07-23

**Typos Grammar Style And Presentation Improvements:** 1. Section 3.2 is not clear, which ne…
**Soundness:** 4

**Excitement:**

4: Strong: This paper deepens the understanding of some phenomenon or lowers the barriers to an existing research direction.

**Missing References:**

[1] Jing, Baoyu, Zeya Wang, and Eric Xing. "Show, Describe and Conclude: On Exploiting the Structure Information of Chest X-ray Reports." Proceedings of the 57th Annual Meeting of the Association for Computational Linguistics. 2019.

[1] is a prior work which also decomposes normal and abnormal reports.

**Paper Topic And Main Contributions:**

This paper introduces a normal-abnormal decoupling memory network for medical report generation. The proposed approach optimizes the visual extraction network only based on anomaly semantics but not normal semantics. The experiments show that the proposed method could outperform baselines.

**Reasons To Accept:**

1. It is interesting to separately model normal and abnormal reports.
2. The proposed approach could outperform baselines on the benchmark MIMIC-CXR dataset.
3. The ablation study shows that each of the component could help improve the performance.

**Reasons To Reject:**

1. The abnormal mode memory part is not quite clear, which is difficult to interpret.
2. The proposed method is somewhat incremental.
3. This paper only includes results of one dataset in the main text.

**Reproducibility:**

4: Could mostly reproduce the results, but there may be some variation because of sample variance or minor variations in their interpretation of the protocol or method.

**Reviewer Confidence:**

4: Quite sure. I tried to check the important points carefully. It's unlikely, though conceivable, that I missed something that should affect my ratings.

---

> ### Author Rebuttal · Authors · 2023-08-27
>
> We sincerely thank you for carefully reviewing our paper, and we offer clarification to your questions.
>
> Q1: The proposed method is somewhat incremental.
>
> We think that our work is novel and relevant. The description of normal varies considerably from doctor to doctor report. To the best of our information, we are the first article to make this noise explicit and attempt to address it. The detrimental impact of this noise is effectively demonstrated in Experiment A.3.3, as elaborated in the Appendix. Our novel proposition of employing the concept of decoupling stands as a logical and effective means to circumvent the influence of this noise. While prior research has explored the separation of normal exceptions to generate reports, it's essential to emphasize that our approach significantly differs in both intent and methodology. As in [1], the data bias problem is considered, where the noise of normal statements still affects the optimization of the model despite the separate generation of normal/abnormal statements. We decouple normal and abnormal semantics, normal semantics are generated independently and gradients do not affect visual optimization. Secondly grouping the reports for optimization emphasizes the role of key semantics rather than unimportant noise. We added additional experiments to illustrate the superiority of our approach. We constructed a separate model of the decoupled structure, with the rest of the modules using the normal Transformer structure, and the hyperparameters remaining the same as in our original model. As shown in Table 1, we find that even without using any other improvements, our method can achieve comparable performance to the previous SOTA model just by utilizing the decoupled approach for training. This proves the significance of our method. Moreover, we believe that this decoupling can be equally inspired by large-scale data. And our decoupled structure can be used in parallel with other improvements without conflict.
>
> Q2: the abnormal mode memory part is not quite clear.
>
> Due to data bias, normal images dominate in the data. We mitigate this problem by using abnormal pattern memory to explicitly memorize abnormal regions of an image. Specifically, inspired by VQ-VAE [2], a discrete memory matrix is created, assuming $N\times D$. The image is encoded by ResNet to obtain the local information of $V \times D$. When memorizing, we look for a unique pattern in the memory space corresponding to each image encoding ($1 \times D$). The Hungarian algorithm [3] is used to find a unique pattern match in the memory matrix by treating the image features with each pattern of the memory matrix as an ensemble matching problem. We utilized orthogonality as well as the Hungarian algorithm [3] in order to find as much information about different patterns as possible (Figure 5 shows that we learned different patterns).  And due to gradient truncation, our memory matrix does not affect visual optimization. We will improve the description of this section in the revision to make it clearer.
>
> Q3: This paper only includes results of one dataset in the main text.
>
> Due to space limitation, we only use the MIMIC-CXR dataset in the main text, but in Appendix A.3.4 we also report the performance of the IU-Xray dataset, which also achieves SOTA. The MIMIC-CXR has a much larger amount of data, and the results are more responsive to the performance of the model.
>
> Q4: Missing References. Misspelled format.
>
> Thanks for pointing out the missing definitions in the article, we will correct the spelling errors as well as clarify the definitions in the revision, where $V_l$  stands for localized image encoding, $V_g$  refers to global image encoding, and k refers to the prediction scores ranked in the first k. And we will add the missing references [1].
>
> Table 1 : Comparison of our simple decoupling model to previous SOTA models on MIMIC-CXR.
>
> | Model    | BLEU-1    | BLEU-2    | BLEU-3    | BLEU-4    | METEOR    | ROUGE-L   |
> | -------- | --------- | --------- | --------- | --------- | --------- | --------- |
> | MSAT [4] | 0.373     | 0.235     | 0.162     | **0.120** | 0.143     | **0.282** |
> | Ours     | **0.389** | **0.241** | **0.163** | 0.117     | **0.147** | 0.281     |
>
> [1] Baoyu Jing, Zeya Wang, and Eric Xing. 2019. Show, describe and conclude: On exploiting the structure information of chest X-ray reports. In Proceedings of the 57th Annual Meeting of the Association for Computational Linguistics, pages 6570–6580, Florence, Italy. Association for Computational Linguistics.
>
> [2] Aaron Van Den Oord, Oriol Vinyals, et al 2017. Neural discrete representation learning. Advances in neural information processing systems, 30.
>
> [3] Harold W Kuhn. 1955. The hungarian method for the assignment problem. Naval research logistics quarterly, 2(1-2):83–97.
>
> [4] Zhanyu Wang, Mingkang Tang, Lei Wang, Xiu Li, and Luping Zhou. 2022c. A medical semantic-assisted transformer for radiographic report generation. In Medical Image Computing and Computer Assisted Intervention – MICCAI 2022: 25th International Conference, Singapore, September 18–22, 2022, Proceedings, Part III, page 655–664, Berlin, Heidelberg. Springer-Verlag.

---

### Official Review · Reviewer_dM6t · 2023-08-04

**Typos Grammar Style And Presentation Improvements:** 1. Line 203, "Combined with global fe…
**Soundness:** 3

**Excitement:**

4: Strong: This paper deepens the understanding of some phenomenon or lowers the barriers to an existing research direction.

**Paper Topic And Main Contributions:**

This work aims to deal with the problem of medical report generation. The develop a visual encoder based on anomaly pattern memory to enhance anomaly perception. The performance shown on MIMIC-CXR is promising over multiple SOTA methods.

**Reasons To Accept:**

1. The idea of using the codebook in VQVAE to store various anomaly information seems to be a right way to go.
2. The memory loss in the objective functions is novel and makes sense.
3. The experimental results seem to be very promising. And the baselines are comprehensive.

**Reasons To Reject:**

I didn't find any obvous reason to reject the paper.

**Reproducibility:**

4: Could mostly reproduce the results, but there may be some variation because of sample variance or minor variations in their interpretation of the protocol or method.

**Reviewer Confidence:**

2: Willing to defend my evaluation, but it is fairly likely that I missed some details, didn't understand some central points, or can't be sure about the novelty of the work.

---

> ### Author Rebuttal · Authors · 2023-08-27
>
> We thank the reviewers for reviewing the article and pointing out the spelling errors, which we will correct in the revision. Our primary objective is to tackle the challenge posed by noise in the domain of report generation. To the best of our information, we are the first article to make this noise explicit and attempt to address it. In our pursuit, we present a novel approach that effectively mitigates the impact of this ambiguous noise by decoupling normal abnormal semantics. Additionally, we introduce an encoder for abnormal pattern memory to heighten the model's sensitivity to anomalies. The performance of our model substantially outperforms the previous SOTA. Hopefully, our research can contribute to the development of automated assisted diagnosis in medicine. Thanks again for reviewing the manuscript!

---

### Official Review · Reviewer_6Vvb · 2023-08-15

**Typos Grammar Style And Presentation Improvements:** 1. The paper focuses on generating th…
**Soundness:** 3

**Ethical Concerns:**

Yes

**Excitement:**

3: Ambivalent: It has merits (e.g., it reports state-of-the-art results, the idea is nice), but there are key weaknesses (e.g., it describes incremental work), and it can significantly benefit from another round of revision. However, I won't object to accepting it if my co-reviewers champion it.

**Justification For Ethical Concerns:**

The paper does not have an ethics section, and since this pertains to a medical application, it is prudent to include a section detailing failures and biases of the model.

**Missing References:**

Please cite Hungarian algorithm in line 221. Please cite the language model that has been used.

**Paper Topic And Main Contributions:**

The paper introduces a novel concept to automatically generate radiology reports from radiographic images using normal-abnormal semantic decoupling network with abnormal pattern memory. The authors combine the concepts of abnormal semantics in reports with visual feature extraction to shape the decoder output. Their trained model achieves higher performance than state of the art models in all of the relevant metrics. In addition, the paper clearly analyses the model output by using model visualization which helps in understanding the model better. The writing is crisp, easy to understand and clearly explains the methodologies used in the paper without ambiguity.

**Questions For The Authors:**

1. What are the details of GPUs used in the experiment and the number of epochs for training, and GPU hours?
2. The paper mentions in line 51 that the presence of noise in reports impacts network optimization, and similar mentions of noise has been repeated in the paper a few times. Please elucidate the type of noise being referred to, and how it impacts network optimization
3. Is the paper using only the Finding section of the radiology report throughout the experiment? It is a bit unclear from the description.
4. Which language model has been used for embeddings for training this model?

**Reasons To Accept:**

1. The methodology achieves higher than state-of-the-art for MIMIC CXR report generation.
2. The methodology is novel, and involves an understanding the structure of radiology report findings beyond just caption generation problem for radiographic images. In radiology reports, the radiologists tend to write more about the abnormalities than focus on the normal perceptions in the report, and this architecture utilizes that aspect in their modeling quite well.
3. The analysis of the model is rigorous and gives us a clear understanding of the types of errors to expect.

**Reasons To Reject:**

1. Though one of the core concepts of the paper is normal-abnormal semantics in reports, the classification from the reports is largely dependent on keyword-based method, which, the authors accept is largely noisy. The definition of what is an abnormal semantics in a report varies depending on application, and there is no specification of any such definition followed in this paper. This makes the normal/abnormal semantic section a little weak compared to the other sections.
-- After rebuttal: Even with the rebuttal, I think this portion could have been handled much better.

**Reproducibility:**

4: Could mostly reproduce the results, but there may be some variation because of sample variance or minor variations in their interpretation of the protocol or method.

**Reviewer Confidence:**

5: Positive that my evaluation is correct. I read the paper very carefully and I am very familiar with related work.

---

> ### Author Rebuttal · Authors · 2023-08-27
>
> Thank you so much for thoroughly reviewing our paper. I greatly appreciate your attention to detail. I will promptly address any questions you have and augment the content accordingly.
>
> Q1: the type of noise and normal-abnormal semantics concepts.
>
> We would like to clarify that the noise sources mentioned in the paper encompass two aspects. Firstly, variations in different descriptions of the same normal semantics . Secondly, differences in reporting habits among various doctors can lead to discrepancies in normal descriptions within reports, where some might be mentioned while others are omitted. Importantly, the semantics themselves are inherently noise-free, as we attribute correctness to the physician's expert judgment. The explicit classification of normal and abnormal semantics is outlined in detail in Appendix A.1. To extract anatomical entities with observations, we employ RadGraph for semantic extraction. For the categorization of normal and abnormal semantics, we rely on using easily detectable normal keywords (e.g., "normal, clear"), as they offer a straightforward means of classification. These keywords are non-ambiguous, limited in scope, and remain relatively consistent. Statements that do not match normal descriptions are presumed to pertain to anomalies, which has demonstrated utility in previous research [1]. We intend to provide comprehensive coverage of this clarification within the main text of the paper.
>
> Q2: how noise impacts network optimization?
>
> In Q1, we clarify the concept of noise. We contend that noise contributes to an element of ambiguity during the training process. The normal description generated by the model may be correct, but omitted from its report (or a different description with the same meaning), while it may be mentioned again in other reports. Our assertion is that this inherent ambiguity disrupts the training process. This interpretation aligns with the outcomes of our noise-related experiments presented in Appendix A.3.3. Specifically, these experiments demonstrate performance enhancements when training gradients that encompass normal semantics containing noise or when training gradients derived from reports with noise are eliminated.
>
> Q3: our motivation for developing the model architecture.
>
> We would like to clarify the motivation of our decoupling structure; as illustrated in Q2, the ambiguity as well as the uncertainty of the normal description can disrupt the training of the model, so we designed this decoupling structure to isolate the interference of this noise. In fact, anomalies are often valued more than normal descriptions in reality. Our decoupling structure realizes this well. We added a separate set of experiments with decoupled structures in our response to other reviewers, and it is exciting to note that without adding any other improvements, our model has reached the level of past SOTA models.
>
> Q4: the types of hallucinations the model and how it differs in its outputs and hallucinations from other state of the art models.
>
> When it comes to modeling hallucinations, our approach is designed to mitigate the impact of noise. This is achieved by decoupling normal abnormal and optimizing report generation for these categories separately. This strategic division enhances the model's sensitivity to anomalies while reducing the prominence of normal descriptions. It is important to acknowledge that our model might produce normal descriptions that deviate from true reports. This could stem from the potential for normal descriptions to be confounded within the training data. Nevertheless, in a real-world context, a primary emphasis should rightfully be placed on anomalies. Misidentified anomalies have a more pronounced adverse impact than redundant normal descriptions.Thanks to the integration of abnormal pattern memory and the model's inherent focus on abnormal semantics, our approach achieves greater score in describing certain anomalies compared to earlier models (our F1-RadGraph and clinical efficiency metrics performs better ). For example, "Consolidation" was ignored in the reports generated by previous models, while our approach mentions this abnormality. However, there are mistakes in the description of some orientations, e.g., left and right are sometimes confused, which is a challenge for generating accurate medical reports. We will add comparisons of our model with other models in the revision to show more examples of the hallucination.
>
> Q5: more detailed evaluation. For example the distribution of diseases in the dataset, the detailed AUC/F1 per disease,  F1-RadGraph,  metrics of text generation like fluency, coherence.
>
> In our study, we opted for common NLG metrics (BLEU, METEOR, ROUGE). Because we are comparing the whole report as an output to the real report, rather than a single sentence, as elaborated in previous work these metrics can to some extent respond to the fluency and coherence [2,3]. Furthermore, to comprehensively demonstrate the performance of our model, we employed the RadGraph-F1 test model, in line with suggestions, and provided a detailed breakdown of disease-specific F1 scores and proportions.  The results are presented in Table 1. It's worth noting that due to the nature of our testing methodology, which involves comparing the label extraction of generated reports, accurate calculation of AUC metrics isn't feasible. Consequently, we exclusively report F1 metrics for each disease in this context. Table 1 represents the RadGraph-F1 scores, where [4,5] are obtained by calculations using the officially provided model weights and [6] are from the original paper. Our metrics achieve the highest scores.  Table 2 represents the F1 scores for the 14 diseases. The weight of this disease occupying all the MIMIC-CXR data is labeled after each category. [4,5] obtained by calculations using the officially provided model weights. It can be seen that the best scores are achieved in most categories. And the percentage of anomalies is small in terms of category share. Detection failures can be due to different reasons, such as too little training data (e.g., "fracture", "Lesion") or too difficult to learn (e.g., "pneumothorax", which is is also difficult for clinicians to determine).
>
> Q6: details about the language model used for embeddings.
>
> We did not use an existing language model for embedding, but only used a general embedding layer for language embedding, which will be trained as the model is trained. We used to use Bio_ClinicalBERT [7] for embedding in the early stage, but there was not much difference in its performance. So we use general embbedding to facilitate the training together.
>
> Q7: what are the details of GPUs used in the experiment and the number of epochs for training, and GPU hours? Missing References. Misspelled format.
>
> We apologize for not being explicit, we used 2080ti at MIMIC-CXR for 30 epoch of training for a total of 26 GPU hours. Thanks for pointing out the spelling errors in the article, we will correct the spelling errors and missing definitions and add the missing references [8] in the revision. We will also make it clear in the text that we are focusing on the findings section of the report.
>
> Q8: The paper does not have an ethics section,it is prudent to include a section detailing failures and biases of the model.
>
> We wish to clarify and emphasize our ethical considerations in this context. The MIMIC-CXR datasets employed in our study underwent a meticulous de-identification process. Our utilization of the MIMIC-CXR dataset aligns with the PhysioNet Health Data license 1.5.0 7. Importantly, the distributed MIMIC-CXR dataset underwent thorough processing to remove all instances of protected health information (PHI). This approach underscores our commitment to adhering to ethical standards. Secondly we will add the failure and bias part of the model further to limitation, in fact, data bias is the main cause of failure and bias. For example, fracture accounts for less than 2% of the entire dataset. In the test, it is difficult for the model to detect report the case of fracture. Secondly, the description ''acute oblique translucency through the right posterior twelfth rib'', which is specific to the first rib, could not be given or was reported incorrectly. We will add more negative sample analyses.
>
> Table 1 : Comparison of our model to previous report generation models on MIMIC-CXR.
>
> | Model        | RadGraph-F1 |
> | ------------ | ----------- |
> | R2Gen [4]    | 0.165       |
> | R2GenCMN [5] | 0.182       |
> | X-REM [6]    | 0.181       |
> | Ours         | **0.209**   |
>
> Table 2 : The detailed results across all CheXpert categories for previous report generation models and our proposed model based on MIMIC-CXR.
>
> | Class(%)                             | R2Gen | R2GenCMN  | Ours      |
> | ------------------------------------ | ----- | --------- | --------- |
> | No Finding (33.119%)                 | 0.512 | **0.513** | 0.471     |
> | Enlarged Cardiomediastinum  (7.266%) | 0.360 | 0.335     | **0.344** |
> | Cardiomegaly (22.336%)               | 0.377 | 0.405     | **0.406** |
> | Lung Lesion (3.259%)                 | 0.329 | **0.327** | 0.322     |
> | Lung Opacity (24.297%)               | 0.350 | 0.350      | **0.364** |
> | Edema (17.641%)                      | 0.385 | 0.428     | **0.429** |
> | Consolidation (6.632%)               | 0.335 | 0.354     | **0.370** |
> | Pneumonia (15.295%)                  | 0.347 | 0.350     | **0.355** |
> | Atelectasis (24.639%)                | 0.369 | **0.398** | 0.366     |
> | Pneumothorax (5.044%)                | 0.338 | 0.339     | **0.348** |
> | Pleural Effusion (26.386%)           | 0.383 | 0.463     | **0.464** |
> | Pleural Other (1.218%)               | 0.326 | **0.331** | **0.331** |
> | Fracture (2.171%)                    | 0.322 | 0.322     | **0.328** |
> | Support Devices (29.318%)            | 0.506 | **0.522** | 0.507     |
>
> [1] Ke Yu, Shantanu Ghosh, Zhexiong Liu, Christopher Deible, and Kayhan Batmanghelich. 2022. Anatomyguided weakly-supervised abnormality localization in chest x-rays. In Medical Image Computing and Computer Assisted Intervention – MICCAI 2022, pages 658–668, Cham. Springer Nature Switzerland.
>
> [2] Hoang Nguyen, Dong Nie, Taivanbat Badamdorj, Yujie Liu, Yingying Zhu, Jason Truong, and Li Cheng. 2021. Automated generation of accurate & fluent medical X-ray reports. In Proceedings of the 2021 Conference on Empirical Methods in Natural Language Processing, pages 3552–3569, Online and Punta Cana, Dominican Republic. Association for Computational Linguistics.
>
> [3] Justin Lovelace and Bobak Mortazavi. 2020. Learning to generate clinically coherent chest X-ray reports. In Findings of the Association for Computational Linguistics: EMNLP 2020, pages 1235–1243, Online. Association for Computational Linguistics.
>
> [4] Zhihong Chen, Yan Song, Tsung-Hui Chang, and Xiang Wan. 2020. Generating radiology reports via memory-driven transformer. In Proceedings of the 2020 Conference on Empirical Methods in Natural Language Processing (EMNLP), pages 1439–01449, Online. Association for Computational Linguistics.
>
> [5] Zhihong Chen, Yaling Shen, Yan Song, and Xiang Wan. 2021b. Cross-modal memory networks for radiology report generation. In Proceedings of the 59th Annual Meeting of the Association for Computational Linguistics and the 11th International Joint Conference on Natural Language Processing (Volume 1: Long Papers), pages 5904–5914, Online. Association for Computational Linguistics.
>
> [6] Jaehwan Jeong, Katherine Tian, Andrew Li, Sina Hartung, Subathra Adithan, Fardad Behzadi, Juan Calle, David Osayande, Michael Pohlen, and Pranav Rajpurkar. 2023. Multimodal image-text matching improves retrieval-based chest x-ray report generation. In Medical Imaging with Deep Learning.
>
> [7] Emily Alsentzer, John Murphy, William Boag, WeiHung Weng, Di Jin, Tristan Naumann, and Matthew McDermott. 2019. Publicly available clinical BERT embeddings. In Proceedings of the 2nd Clinical Natural Language Processing Workshop, pages 72–78, Minneapolis, Minnesota, USA. Association for Computational Linguistics.
>
> [8] Harold W Kuhn. 1955. The hungarian method for the assignment problem. Naval research logistics quarterly, 2(1-2):83–97.

---

### Meta-Review · Area_Chair_zjmh · 2023-09-19

**Recommendation:** 3

**Metareview:**

The paper investigates the automatic generation of radiology reports from X-ray images (MIMIC-CXR). The authors' approach centers on optimizing the visual encoder based on anomaly semantics, deliberately excluding normal semantics to mitigate the impact of potential noisy signals.

All reviewers acknowledge the effectiveness of the proposed method in the radiology report generation domain, noting improvements relative to current state-of-the-art benchmarks. Comprehensive ablations and analyses are also presented. Some concern is expressed regarding how to determine abnormal vs. normal semantics (a naive heuristic is used which can be noisy itself).

---

### Decision · Program_Chairs · 2023-10-07

**Decision:**

Accept-Findings

**Comment:**

The paper investigates the automatic generation of radiology reports from X-ray images (MIMIC-CXR). The authors' approach centers on optimizing the visual encoder based on anomaly semantics, deliberately excluding normal semantics to mitigate the impact of potential noisy signals.

All reviewers acknowledge the effectiveness of the proposed method in the radiology report generation domain, noting improvements relative to current state-of-the-art benchmarks. Comprehensive ablations and analyses are also presented. Some concern is expressed regarding how to determine abnormal vs. normal semantics (a naive heuristic is used which can be noisy itself).